# Bats and Coronaviruses

**DOI:** 10.3390/v11010041

**Published:** 2019-01-09

**Authors:** Arinjay Banerjee, Kirsten Kulcsar, Vikram Misra, Matthew Frieman, Karen Mossman

**Affiliations:** 1Department of Pathology and Molecular Medicine, Michael DeGroote Institute for Infectious Disease Research, McMaster University, Hamilton, ON L8S 4L8, Canada; banera9@mcmaster.ca; 2Department of Microbiology and Immunology, University of Maryland School of Medicine, Baltimore, MD 21201, USA; KKulcsar@som.umaryland.edu (K.K.); MFrieman@som.umaryland.edu (M.F.); 3Department of Veterinary Microbiology, Western College of Veterinary Medicine, University of Saskatchewan, Saskatoon, SK S7N 5B4, Canada; vikram.misra@usask.ca

**Keywords:** bats, coronaviruses, immune response, in vitro, in vivo

## Abstract

Bats are speculated to be reservoirs of several emerging viruses including coronaviruses (CoVs) that cause serious disease in humans and agricultural animals. These include CoVs that cause severe acute respiratory syndrome (SARS), Middle East respiratory syndrome (MERS), porcine epidemic diarrhea (PED) and severe acute diarrhea syndrome (SADS). Bats that are naturally infected or experimentally infected do not demonstrate clinical signs of disease. These observations have allowed researchers to speculate that bats are the likely reservoirs or ancestral hosts for several CoVs. In this review, we follow the CoV outbreaks that are speculated to have originated in bats. We review studies that have allowed researchers to identify unique adaptation in bats that may allow them to harbor CoVs without severe disease. We speculate about future studies that are critical to identify how bats can harbor multiple strains of CoVs and factors that enable these viruses to “jump” from bats to other mammals. We hope that this review will enable readers to identify gaps in knowledge that currently exist and initiate a dialogue amongst bat researchers to share resources to overcome present limitations.

## 1. Introduction

Bats are an ancient and diverse group of ecologically important mammals, constituting almost a quarter of all mammalian diversity and inhabiting every continent except Antarctica. More than 1300 species of bats belong to the order Chiroptera and are further classified into two suborders, Yinpterochiroptera and Yangochiroptera [1,2,3]. The Yinpterochiroptera suborder includes the non-echolocating Pteropodidae family and the echolocating Rhinolophoidea superfamily. Yangochiroptera contains the remaining echolocating microbat families. The two suborders diverged over 50 million years ago [4,5,6]. In addition to the important role that bats play in preservation of ecological balance, they have also been speculated to harbor a wide variety of viruses. Many of the viruses in bats can cause disease in humans and agriculturally important animal species. These viruses include lyssaviruses, filoviruses, henipaviruses and coronaviruses [1,7,8,9].

In this mini-review, we focus on the role of bats as reservoir hosts for important human and animal coronaviruses. We discuss the evidence of coronavirus spillover from bats, how bat ecological niches may contribute to spillover and the need to further explore bat-coronavirus interactions using viruses that have been naturally detected in bats.

## 2. Coronaviruses and Their Origins

The coronavirus family (*Coronaviridae*) is the largest in the order *Nidovirales*. Coronaviridae consists of two subfamilies, Letovirinae and Orthocoronavirinae. Within the subfamily Orthocoronavirinae, there are four genera: alphacoronavirus, betacoronavirus, gammacoronavirus and deltacoronavirus. Alphacoronaviruses and betacoronaviruses are found in mammals, whereas gammacoronaviruses and deltacoronaviruses are primarily found in birds [10]. Coronaviruses are enveloped and have single stranded positive sense RNA genomes that range in size from 26 to 32 kilobases [11]. The genome encodes a variety of structural, non-structural and accessory proteins. The accessory proteins vary between coronaviruses, even within the same clade, and perform a variety of functions including antagonism of the host response following infection [12,13,14]. Coronaviruses typically cause respiratory and enteric diseases in humans and animals, respectively. Historically, coronaviruses were thought to cause agriculturally important diseases in animals and the common cold in humans. However, in 2002, severe acute respiratory syndrome coronavirus (SARS-CoV) emerged in China. The SARS-CoV outbreak lasted for 8 months and resulted in 8,098 confirmed human cases worldwide, of which 774 (9.5%) were fatal [15]. Approximately 10 years after SARS-CoV emerged, another highly pathogenic human coronavirus, Middle East respiratory syndrome coronavirus (MERS-CoV) appeared in the Kingdom of Saudi Arabia. MERS-CoV continues to cause outbreaks and has resulted in a total of 2,260 confirmed cases in 27 countries of which 803 have been fatal [16]. Currently, there are four coronaviruses that have been recognized to cause the common cold in humans, HCoV-OC43, HCoV-NL63, HCoV-HKU1, and HCoV-229E, and two emerging coronaviruses, SARS-CoV and MERS-CoV, that can cause highly pathogenic respiratory infections [17].

In addition to the human coronaviruses, other coronavirus species cause agriculturally important diseases in animals. These include avian infectious bronchitis virus (IBV), transmissible gastroenteritis virus (TGEV), porcine epidemic diarrhea virus (PEDV) and more recently, swine acute diarrhea syndrome coronavirus (SADS-CoV). PEDV led to a 3.21% decrease in the U.S. pig crop between September 2012 and August 2014 [18] and IBV is estimated to cause $3567.40 in losses per 1000 birds on poultry farms in Brazil [19].

Many of these human and animal coronaviruses appear to have origins in a variety of bat species. With the advent of next generation sequencing technology and increased surveillance of wild animal species, an enormous number of novel coronaviruses have been identified. To date, over 200 novel coronaviruses have been identified in bats and approximately 35% of the bat virome sequenced to date is composed of coronaviruses [20]. These technologies have helped elucidate the evolutionary history of coronaviruses that are known to cause disease in humans and animals of agricultural importance (Table 1).

### 2.1. Human Coronavirus Origins

SARS-CoV emerged in humans in 2002 and efficient human to human transmission resulted in a global SARS epidemic which lasted 8 months [15]. Initial studies investigating animal sources of the virus from “wet markets” in the Guangdong province of China suggested that Himalayan palm civets and raccoon dogs were the most likely hosts responsible for human transmission [22]; however, the role of bats as the original animal reservoir hosts of SARS-CoV was speculated as similar viruses were detected in them [27,28]. Years later, during an ecological surveillance of bats in the same region, a SARS-like CoV that closely matched the human SARS-CoV was isolated from the Chinese horseshoe bat. Bat SARS-like CoV could replicate in HeLa cells expressing angiotensin-converting enzyme 2 (ACE2) receptor from human, civet and bat. The virus replicated in cells derived from human, bat and pig. No civet cells were tested [29]. These data suggest that SARS-CoV could have spilled over into humans directly from the Chinese horseshoe bat while the palm civets in the “wet market” were incidental hosts [30]. However, the exact mechanism by which the zoonotic transmission event to humans occurred is still not clear. Retrospective studies have found low levels of seroprevalence of SARS-like CoV in healthy individuals in Hong Kong dating back to 2001. Interestingly, in 13 of 17 of these seropositive patients, the antibodies responded more strongly against the SARS-like-CoV isolated from a Himalayan palm civet compared to the human SARS-CoV isolate [31]. These data suggest that low levels of human exposure to zoonotic SARS-like CoVs occurred prior to the SARS-CoV epidemic that began in 2002, but went unidentified.

MERS-CoV emerged in Saudi Arabia in 2012 and continues to cause human disease with a case fatality rate of 35% [16]. Dromedary camels are a natural reservoir host for MERS-CoV. In the Arabian Peninsula and across Northern Africa, the seroprevalence rate for MERS-CoV in dromedary camels ranges from 70% to nearly 100% [32,33,34,35,36,37]. Live MERS-CoV has been successfully isolated and cultured from camel specimens [38]. Approximately 55% of primary MERS-CoV cases are a result of direct contact with dromedary camels or camel products [39]; however, the remainder of primary MERS-CoV cases have no history of contact with camels or infected individuals and thus, where they came into contact with the virus is unknown. A recent study found that 16 out of 30 camel workers surveyed in Saudi Arabia show evidence of prior MERS-CoV infection via seroconversion and/or virus-specific CD8+ T cell responses without any history of significant respiratory disease. This study suggests that camel workers with asymptomatic or mild disease may serve as another route of exposure [40]. Although camels are thought to be the primary zoonotic reservoir for human transmission, there is strong evidence that bats are the ancestral reservoir host for MERS-CoV [24,41,42,43]. MERS-CoV is a group C betacoronavirus and is phylogenetically related to BatCoVs identified in various bat species that belong to the Vespertilionidae family. This includes BatCoV HKU4, BatCoV HKU5, NeoCoV, and PDF-2180 [41,44,45]. Furthermore, the spike protein from HKU4 and MERS-CoV are highly similar and both use human dipeptidyl-peptidase 4 (DPP4) for virus entry [25,46,47]. It is not clear when MERS-CoV spread from bats to camels, but widespread exposure to the virus in the Middle East and North and East Africa dates back as early as the 1980s, suggesting that camels have served as a zoonotic reservoir for MERS-CoV for at least 30 years [33,34,37,40,48].

In addition to emerging highly pathogenic coronaviruses, human coronaviruses that cause the common cold are also thought to have their origins in bats. HCoV-NL63 was first identified in a pediatric patient with bronchiolitis in 2004, but since then it has come to be appreciated that the virus causes approximately 1–9% of the common colds each year and has most likely circulated in humans for centuries with worldwide distribution [49,50,51]. A survey of bats in the U.S. found novel alphacoronaviruses, one of which was isolated from the North American tricoloured bat, *Perimyotis subflavus* and was found to be a recent common ancestor of HCoV-NL63 with an estimated divergence of ~550 years ago [52]. HCoV-NL63-like sequences were also identified in bats in Africa [53], further supporting a bat origin for HCoV-NL63. Although HCoV-NL63-like viruses have been identified in bats, these viruses have sequences quite distant from the HCoV-NL63 sequences, suggesting a possible intermediate host. HCoV-229E also appears to have its origins in bat species. HCoV-229E, another cause of the common cold, was first identified in 1967 and has been circulating in the human population for some time [54]. HCoV-229E-related viruses have been found in hipposiderid bats during surveillance studies in Kenya and Ghana [53,55]. In 2007, a novel alphacoronavirus was identified in an outbreak of respiratory disease in alpacas in the US, which is geographically separated from the bat species that harbor HCoV-229E-like viruses in Africa [56]. Full genome sequencing and phylogenetic analysis of the alpaca CoV placed it as an intermediate between the bat HCoV-229E-related viruses and HCoV-229E from humans [56]. By analyzing more bat, alpaca and human HCoV-229E and HCoV-229E-related sequences, evidence of genomic changes that occurred between bat and alpaca HCoV-229E evolution and subsequently between alpaca and human evolution were identified [57]. Interestingly, during tests of dromedary camels for MERS-CoV, about 6% of the camels studied were positive for HCoV-229E [58]. Seropositive camels were more prevalent in the Arabian Peninsula compared to Africa and the earliest seropositive sample was from 1997 in a study that looked at samples from 1983 to 2014 [58]. These data all support the notion that HCoV-229E has its ancestral origins in bat species while camelids serve as a more recent zoonotic reservoir for human transmission. A recent study has shown that HCoV-229E (human strain) is incapable of infecting and replicating in cell lines from multiple bat species [59]. Thus it is critical to isolate bat and camel strains of HCoV-229E-related viruses to dissect the role of these mammals in the evolution of HCoV-229E.

### 2.2. Animal Coronavirus Origins

Porcine epidemic diarrhea (PED) was recognized as an enteric disease in pigs in the United Kingdom as early as 1971. PEDV was detected in Belgium in 1978 [60]. The full-length genomic sequence of the prototype Belgian PEDV CV777 strain was determined in 2001 [61]. PEDV CV777 is more closely related to a *Scotophilus* bat coronavirus (BtCoV) 512/2005 than to other known alphacoronaviruses, such as transmissible gastroenteritis coronavirus (TGEV) and HCoV-229E and HCoV-NL63, in phylogeny as well as genome organization [21]. This finding suggests that PEDV and BtCoV/512/2005 have a common evolutionary precursor and that cross-species transmission of coronavirus may have occurred between bats and pigs. PEDV has since emerged in North America and continues to cause periodic outbreaks that significantly affect producers [18,62]. Multiple PEDV vaccine candidates have been shown to provide varying levels of protection in pigs [63,64]. An effective vaccine may enable control of future PEDV outbreaks along with strict biosecurity practices. Although PEDV propagates in human embryonic kidney cells [65], no clinical cases of PEDV have been reported in humans so far. We (Banerjee and Misra et al.) have also shown that PEDV can infect kidney cells from big brown bats (*Eptesicus fuscus*) [66]. PEDV replication in bat cells has not been extensively studied. Efforts are focused on designing therapeutics and vaccines to prevent PED in pigs.

More recently, a novel HKU2-related bat coronavirus, SADS-CoV has been shown to cause fatal enteric disease in pigs. Zhou et al. identified SADS-CoV as the causative agent for a large-scale outbreak of fatal disease in pigs in China that caused the death of 24,693 piglets across four farms. SADS-CoV-like viruses with 96–98% similarity to SADS-CoV were detected in 9.8% of the bats that were sampled in this region [9]. None of the human serum samples that were collected from farm workers were positive for antibodies against SADS-CoV [9]. Thus, SADS-CoV does not pose a risk for human transmission yet. Further studies will be required to confirm the ability of SADS-CoV to infect and propagate in human cells.

## 3. Bats and Coronavirus Spillover Events

Understanding how bats maintain a virus within a population is important for predicting spillover transmission events. For many viruses with known or suspected bat reservoirs, spillover transmission events typically occur within a defined time frame and location, which corresponds with higher than normal virus levels in the bat reservoir host. The reason for these “pulses” of virus within the reservoir host population are not clear, but proposed theories and evidence supporting these theories have been reviewed by Plowright et al. [67]. In the case of Marburg virus (MARV), for example, ecological surveillance data shows a clear biannual spike in the prevalence of MARV positive bats within the Kitaka cave population, which correlates with an increase in the number of juvenile *Rousettus aegyptiacus* bats due to the biannual birthing cycle. This pulse of virus positive bats correlates with an increased incidence of human spillover events [68]. Horizontal transmission of MARV between *R. aegyptiacus* bats was confirmed in a controlled experimental setting [69]. Furthermore, recent experimental data has shown that bats infected with MARV clear infection and maintain long-term immunity. This finding suggests that susceptible naïve juvenile bats are critical for maintaining MARV within the population [69].

Studies with Hendra virus have shown that reproductive and nutritional stress can increase the levels of virus in little red flying foxes (*Pteropus scapulatus*) [70]. The increase in virus replication may enhance the chances of a virus spillover. Similar ecological studies need to be undertaken for bats and CoVs. Other stressors, such as secondary infections, may also affect the relationship between bats and their viruses. A recent study by one of our laboratories (Misra et al.) suggest that infection of little brown bats (*Myotis lucifugus*) with White-Nose Syndrome causing fungus (*Pseudogymnoascus destructans*) leads to an increase in replication of a persistently infecting coronavirus in these bats [71].

A recent study by Anthony et al. evaluated the global diversity of coronaviruses in almost 20,000 animals and humans. During the course of this study, they found that the diversity of coronaviruses was highly associated with the diversity of bat species and this diversity separated into 3 distinct geographical regions, which mirrored the distribution of different species of bats. The authors report particular associations between bat families and viral sub-clades that suggest co-evolution [72]. A survey of coronaviruses isolated from bats in Kenya found a high prevalence of coronaviruses in *Cardioderma cor*, *Ediolon helvum*, *Epomophorus labiatuc*, *Hipposideros* sp., *Miniopterus minor*, *Otomops martiensseni*, *Rhinolophus hildebrandtii*, *Rhinolophus* sp., and *Triaenops afer*. The phylogenetic analysis of these novel CoVs found a number of cross-species transmission events, although the majority of these events appeared to be transient spillover events [53]. The recombination frequency of coronaviruses, which can be as high as 25% for the entire genome [73], could lead to bats being an important reservoir for coronavirus recombination and virus evolution, much like birds and pigs are for influenza virus. Indeed, there is strong evidence to suggest that a recombination event occurred between HCoV-229E-like viruses found in *Hipposideros* bats and HCoV-NL63-like viruses found in *Triaenops afer* bats, where the gene encoding for the spike protein is more closely related to the HCoV-229E virus [53]. Furthermore, the majority of recombination events identified in coronaviruses isolated from bats suggest recombination hotspots around the spike gene [53,23]. In theory, bats could serve as an important reservoir for coronaviruses and coronaviruses with altered host tropism may very well evolve in bats.

Although bats are known to harbor a wide variety of coronaviruses, the mechanisms for virus spillover into humans or livestock are widely unknown. There is evidence that there are seasonal fluctuations in virus replication [74,75], however, the interconnectedness of virus replication rates and virus spillover have not been explored for bats. Typically, coronaviruses found in bats have or require an intermediate host before spilling over into humans, like what is observed with MERS-CoV and camels. Unlike the amount of information available from studies of other bat viruses such as Nipah, Hendra, Ebola, and Marburg viruses, we know very little, if anything about how coronaviruses are transmitted directly to humans or if direct human transmission does not occur and spillover via an intermediate host is required.

## 4. Bat Immune Response to Coronaviruses

Bats are known to harbor a wide range of viruses including many that are highly pathogenic in humans. Research to determine the mechanisms by which bats limit disease following virus infection is a relatively new field and can be difficult due to a lack of reagents and the need to develop appropriate in vitro and in vivo systems. Even with these limitations, a variety of studies have been performed that evaluate the bat immune response to virus infection at the genomics level, in vitro using cell culture systems, and performing experimental infections in vivo. Of note, very few of these studies are focused on coronavirus infections in bats and are rather centered around henipavirus and filovirus infections. Future studies evaluating the virus-host interactions of bats and coronaviruses, particularly with bat CoV isolates are important in determining why bats serve as important reservoirs for CoVs and how they control infection to limit severe pathological consequences.

### 4.1. Cell Culture Model Systems

Multiple studies have elucidated unique adaptations in the antiviral responses of bat cells. The primary bat species being used to study the bat immune response to virus infections in vitro and in vivo are *Pteropus alecto* (black flying fox), *Rousettus aegyptiacus* (Egyptian rousette), and *Artibeus jamaicensis* (Jamaican fruit bat). Papenfuss et al. were the first to sequence the *P. alecto* transcriptome and identified approximately 500 genes (3.5% of *P. alecto* transcribed genes) that encode immune-related proteins [76]. A similar number of immune genes were also identified in the transcriptomes of *R. aegyptiacus* and *A. jamaicensis* [77,78]. This included the expression of canonical pattern recognition receptors including toll-like receptors (TLRs) 1–10, retinoic acid-inducible gene I (RIG-I), and melanoma differentiation associated protein 5 (MDA5) [76,77]. Furthermore, genes for different immune cell subsets, T-cell receptors (TCRs), cytokines and chemokines, and interferon-related genes were detected, while genes encoding for natural killer (NK) cell receptors were largely absent. Work has been done to characterize many of these genes in cell lines derived from various bat species including *P. alecto* [79,80,81].

A large amount of interest in bat immune responses has focused specifically on the interferon response. Genomic analysis of the interferon loci has shown species-specific evolution in which *P. alecto* has a contracted type I IFN locus [82] whereas *P. vampyrus*, *M. lucifugus*, and *R. aegyptiacus* have expanded the number of type I IFN genes [83,84]. It has been observed that there may also be species specific differences in the baseline expression of type I IFNs. *P. alecto* cells constitutively express three different IFNα genes [82] whereas cells generated from *R. aegyptiacus* do not show constitutive expression of IFNα [84]; however, baseline expression of interferon alpha/beta receptors, *IFNAR1* and *IFNAR2*, as well as a variety of interferon-stimulated genes are upregulated in these bat cells compared to human cells [85]. The molecular mechanisms that enable the differential expression pattern of IFNs in bats are not known. Thus, it is important to acknowledge that different species of bats may have evolved specific strategies to control viruses that they co-evolved with.

Although it appears that bats have many of the genes that are important for responding to virus infection, how this response compares between human and bat cells is just beginning to be examined. RNA sensing and subsequent antiviral responses in bat cells have been studied using viruses known to induce an interferon response, such as Sendai virus or Newcastle disease virus or by transfecting a synthetic surrogate of viral double stranded RNA (poly(I:C)) [80,86,87,88,89,90]. These studies show that bat cells respond to RNA and induce an antiviral response. Many viruses encode proteins that antagonize the host response to infection and dampen the innate antiviral response. It has previously been shown that the V and W proteins of Nipah and Hendra viruses can inhibit antiviral responses in bat cells, similar to what is observed in human cells [90]. A more recent study showed that MARV can inhibit the antiviral response in a *R. aegyptiacus* bat cell line and that this inhibition is dependent on the viral protein VP35 [85].

Coronavirus accessory proteins are dispensable for replication but they play an important role in pathogenesis and virus fitness under the natural environment of a host [13,91]. Multiple studies with PEDV, SARS- and MERS-CoVs have identified accessory proteins that can effectively inhibit an IFN response in mammalian cells [12,13,14,91,92,93,94,95]. However, to date, there have been no published studies looking at the role of these accessory proteins in modulating antiviral responses in bat cells.

In addition to studying the role of CoV proteins in antagonizing the antiviral response in bat cells compared to other mammalian cell lines, it is also important to determine how CoVs isolated from bats compare to those isolated from humans. Coronavirus accessory genes have co-evolved with their natural host for optimum functionality [91] and thus it is important to identify the role of accessory proteins in both their natural and spillover hosts. Many of the CoVs that have been reported in bats, with the exception of few, such as a SARS-like CoV (bat SL-CoV-WIV1) [29], have been detected by molecular techniques that detect trace amounts of viral nucleic acids. To overcome this limitation, reverse genetics systems using the whole genome sequence from CoVs isolated from bats could be generated, propagated and evaluated in both bat and human cell lines [96]. This would allow researchers to better understand the role of viral proteins in a species-specific context.

### 4.2. In vivo Model Systems

The vast majority of studies evaluating the bat host response to virus infection has been performed in cell lines. However, there is a great need to understand what happens during a virus infection in bats in vivo. The ability to study these questions is a daunting task and requires specialized facilities and staff, appropriate species selection especially for CoVs, and generating the necessary reagents. Because of these limitations, only a handful of studies have been performed looking at the in vivo response of bats to virus infection. In fact, there are only two published studies in which experimental infections in bats using CoVs was performed.

The first study was performed in an attempt to rescue a bat CoV isolate. Watanabe et al. detected CoVs in 57.1% of insectivorous bats and 55.6% of frugivorous bats; however, they were unable to culture the virus in vitro. To propagate CoVs detected in a lesser dog-faced fruit bat (*Cynopterus brachyotis*), they administered intestinal samples orally to Leschenault rousette bats (*Rousettus leschenaulti*). Virus could be detected by quantitative real-time PCR (qPCR) on 2 to 5 days after infection and there was an increase in viral RNA while no clinical disease was observed. Based on these data, the authors reported that this bat CoV replicates in Leschenault rousette bats; however, they were not able to isolate live virus [97]. This study emphasizes the importance of bat species selection for studying CoVs in bats. Ideally, we would want to study a bat CoV in the same species that it was detected in.

The second study aimed to determine if bats could be infected with MERS-CoV and, if so, what the host response looks like. Munster et al. infected ten Jamaican fruit bats (*Artibeus jamaicensis*) with MERS-CoV/EMC2012. The authors detected virus shedding in the respiratory and intestinal tracts for 9 days. Although the bats showed evidence of virus replication, no overt signs of disease were observed. A moderate and transient induction of the innate immune response was seen, but there were no signs of inflammation. Based on their observations, the authors reported that Jamaican fruit bats support the replication of MERS-CoV and thus, bats could be potential ancestral hosts of MERS-CoV [98]. This study has not been repeated in an insectivorous bat species. Although several MERS-like viruses have been detected in bats since the study in Jamaican fruit bats, none have been successfully isolated [25,41,47,99].

Another study focused on looking at species-specific tropism. In this study, the authors focused on the MERS-CoV receptor dipeptidyl peptidase 4 (DPP4). Widagdo et al. mapped the tissue distribution of DPP4 in multiple bat species to identify the differences in tissue tropism of MERS-CoV. In their study, the authors report that DPP4 in insectivorous bats is primarily detected in the gastro-intestinal (GI) tract and kidneys, whereas frugivorous bats express DPP4 in the respiratory and GI tracts [43]. Other studies determined that DPP4 expression in camels is primarily in the upper respiratory tract [100] whereas DPP4 expression in humans is highest in the lower respiratory tract [101]. These data suggest that the tissue tropism in bats may be different than that in other mammalian species and that this may dictate the course of disease and disease severity.

## 5. Implications of Bats as Hosts of CoVs

The ability of bats to harbor several different coronaviruses may seem like a mystery, but the same is true for rodents. Although bats harbor more zoonotic viruses per species, rodents harbor a larger total number of zoonotic viruses [102]. After the SARS outbreak, bats have been extensively sampled for coronaviruses and other viruses alike. We may be looking too hard and one may argue that we could find a similar diversity of viruses in other animals if we looked as robustly. Metagenomics has enabled us to identify the broad range of viruses in bats and with time, we will expand this to other hosts of zoonotic viruses. For now, we know that bats are major evolutionary reservoirs and ecological drivers of CoV diversity [72]. We can leverage this knowledge to design studies that will allow us to identify factors that cause CoVs to spillover from bats to other hosts.

A recent study demonstrated that secondary infection with the White-nose syndrome fungus (*Pseudogymnoascus destructans*) increases CoV replication in *M. lucifugus* [71]. This study opens up a new avenue of investigation in infection dynamics. Considering bats harbor multiple viruses, it is necessary to identify the impact these viruses have on each other. How do these viruses modulate the numerous host responses in bats and how does that affect virus replication? Several such questions remain and studies are currently delayed due to the inability to isolate bat-CoVs similar to SARS-CoV, MERS-CoV, PEDV and SADS-CoV.

Ecological and epidemiological studies to identify landscape changes and human practices that could enable a coronavirus to spillover from bats are also necessary [103]. Such studies enabled researchers to decipher the transmission cycle of Nipah virus in Malaysia and Bangladesh, which led to public health policies to change farming practices to control the spillover of Nipah virus from bats [104,105,106].

As the human population expands and societal changes occur, human contact with wildlife will continue to increase. This increases the risk that emerging zoonotic viruses, including CoVs, pose to human and animal health. Surveillance combined with scientific studies to better understand zoonotic CoVs and spillover will enable us to stay a step ahead of the next epidemic.

## Figures and Tables

**Table 1 viruses-11-00041-t001:** Zoonotic coronaviruses that have caused serious disease in humans and agricultural animals.

Coronavirus	Affected Host	Intermediate Host	Potential Reservoir/Ancestral Hosts	Similar Virus in Intermediate Host	Similar Virus in Reservoir Host	Reference
PEDV	Pigs	None identified	Bat (*Scotophilus kuhlii*)	None identified	BtCoV/512/05	[21]
SADS-CoV	Pigs	None identified	Bat (*Rhinolophus* spp.)	None identified	HKU2-CoV	[9]
SARS-CoV	Humans	Himalayan palm civet/racoon	Bat (*Rhinolophus* spp.)	CoV isolate SZ3 and SZ16	SARS-related CoVs	[22,23]
MERS-CoV	Humans	Dromedary camels	Bat (*Taphozous perforatus*, *Rhinopoma hardwickii* and *Pipistrellus kuhlii*)	MERS-CoV—KFU-HKU 1 and KFU-HKU 13	BatCoV Rhhar, BatCoV Pikuh, BatCoV Taper	[24,25,26]

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
