# Peer review of "Bats and Coronaviruses"

_viruses, 2019, doi:10.3390/v11010041_

Reviewer 1 Report

The article reviews the relationship between bats and coronaviruses and thus may increase our understanding of the current knowledge and future studies regarding how bats can harbor multiple coronaviruses and the factors which may enable the cross species infection. The manuscript is an appropriate review article; however, the modifications as suggested below are required in order to increase the clarity for the readers.

Specific commends

1. Line 19, “review” or “introduce” instead of “mention” would be more appropriate.

2.Line 34, “In addition to the important role that bats play” instead of “In addition to the important role bats play” would increase the clarity.

3.Lines 43-44, “Within the subfamily Coronavirinae, there are four clades or genera:” instead of “Within the coronavirus family, there are four clades or subfamilies:” That is, genera (coronavirus), subfamily Coronavirinae, family Coronaviridae (includes subfamilies Coronavirinae and Torovirinae).

4. Line 48, “kilobases” instead of “KB”.

5. Line 46, it would be better to cite a reference at the end of the sentence.

6. Line 51, “Coronaviruses typically cause respiratory and enteric diseases in humans and animals.” would be more appropriate than “Coronaviruses typically cause respiratory disease in humans and enteric disease in animals.”.

7. Line 62, it would be clear to cite a reference at the end of the sentence.

8. Line 64, “ These include” would be more appropriate than “ This includes”.

9. Table 1, Split a word into two or three parts (for example, Coro navir us and Affect ed in the first two panels of the Table 1 and Ref ren ce in the last panel of the table) may decrease the clarity of the table. It is suggested to combine the parts into one word (Coro navir us becomes Coronavirus).

10. Line 82, It would be better to give a rationale or a reason why bats are speculated to be reservoir hosts in order to increase the clarity.

11. Line 101, “and thus” or “. Thus,” instead of “, thus,”

12. Lines 127 and 132, “HCoV-229E-related “ instead of “HCoV-229E related”.

13.Lines 145-146, As far as my understanding (or I may be wrong), PEDV (strain CV777) may not be a devastating enteric pathogen for pigs back in 1970s. Or the authors may cite a reference to support the description.

14. Line 184. Since fungus causing White-nose syndrome is associated with increased replication of a naturally persisting coronaviruses in bats, it would be better to use the term” affect” or other alternatives instead of “disrupt”.

15. Line 186, “ with White-nose syndrome fungus “ instead of “by the White-Nose Syndrome causing fungus” may be more accurate.

Author Response

Reviewer 1:

Specific comments:

1. Line 19, “review” or “introduce” instead of “mention” would be more

appropriate.

Authors’ response: done.

2.Line 34, “In addition to the important role that bats play” instead of “In

addition to the important role bats play” would increase the clarity.

Authors’ response: done.

3.Lines 43-44, “Within the subfamily Coronavirinae, there are four clades or

genera:” instead of “Within the coronavirus family, there are four clades or

subfamilies:” That is, genera (coronavirus), subfamily Coronavirinae, family

Coronaviridae (includes subfamilies Coronavirinae and Torovirinae).

Authors’ response: done. We have included the latest ICTV classification.

4. Line 48, “kilobases” instead of “KB”.

Authors’ response: done.

5. Line 46, it would be better to cite a reference at the end of the sentence.

Authors’ response: done.

6. Line 51, “Coronaviruses typically cause respiratory and enteric diseases in

humans and animals.” would be more appropriate than “Coronaviruses

typically cause respiratory disease in humans and enteric disease in animals.”.

Authors’ response: done.

7. Line 62, it would be clear to cite a reference at the end of the sentence.

Authors’ response: done.

8. Line 64, “ These include” would be more appropriate than “ This includes”.

Authors’ response: done.

9. Table 1, Split a word into two or three parts (for example, Coro navir us and

Affect ed in the first two panels of the Table 1 and Ref ren ce in the last panel

of the table) may decrease the clarity of the table. It is suggested to combine

the parts into one word (Coro navir us becomes Coronavirus).

Authors’ response: we agree with this. Will this be addressed at the stage of typesetting?

10. Line 82, It would be better to give a rationale or a reason why bats are

speculated to be reservoir hosts in order to increase the clarity.

Authors’ response: done.

11. Line 101, “and thus” or “. Thus,” instead of “, thus,”

Authors’ response: replaced it to ‘and thus’.

12. Lines 127 and 132, “HCoV-229E-related “ instead of “HCoV-229E related”.

Authors’ response: done.

13.Lines 145-146, As far as my understanding (or I may be wrong), PEDV

(strain CV777) may not be a devastating enteric pathogen for pigs back in

1970s. Or the authors may cite a reference to support the description.

Authors’ response: we have removed ‘devastating’.

14. Line 184. Since fungus causing White-nose syndrome is associated with

increased replication of a naturally persisting coronaviruses in bats, it would be

better to use the term” affect” or other alternatives instead of “disrupt”.

Authors’ response: we have switched ‘disrupt’ with ‘affect’.

15. Line 186, “ with White-nose syndrome fungus “ instead of “by the White-

Nose Syndrome causing fungus” may be more accurate.

Authors’ response: done.

Reviewer 2 Report

The manuscript give update information regarding the bat and coronaviruses and emphasize particularly on some improtant coronaviruses, inludging the origins, spillover events, bat immnure response to coronavirus infection. This manuscript was concise and well written and should have a broad readership.

Minor comments:

table 1 and line 110, should include the following references: Lau et al., J. Infect Dis., 2018:197-2017; Luo et al., J Virol. 2018 (13), 10.1128/JVI.00116-18

lines 85-86: This description needs to be modified. Bat SARS-like CoV could replicate in HeLa cells expressing ACE2 from human, civet and bat. The virus replicates in cells derived from human, bat and pig. No civet cells were tested.

In the section "bats and coronavirus spillover events", should discuss more on virus property (genetic diversity), ecological factors (seasonal prevalence, intermediate hosts), etc.

line 204, should include the following reference: Hu et al., PLoS pathogens, 2017: e1006698.

Lines 207-, should discuss variation of the viral loads in different season may also contribute the spillover.  Please see the references: Wang et al., Virol Sin, 2016, 31:78-80; Lau et al., J Virol, 2010, 84: 2808-19.

Line 275, should included the following reference: Necker et al., PNAS, 2008, 105:19944-9.

Author Response

Reviewer 2:©

Minor comments:

table 1 and line 110, should include the following references: Lau et al., J.

Infect Dis., 2018:197-2017; Luo et al., J Virol. 2018 (13), 10.1128/JVI.00116-18

Authors’ response: done.

lines 85-86: This description needs to be modified. Bat SARS-like CoV could

replicate in HeLa cells expressing ACE2 from human, civet and bat. The virus

replicates in cells derived from human, bat and pig. No civet cells were tested.

Authors’ response: done.

In the section "bats and coronavirus spillover events", should discuss more on

virus property (genetic diversity), ecological factors (seasonal prevalence,

intermediate hosts), etc.

Authors’ response: We have moved the paragraph on ‘Hendra virus and the role of nutritional stress in virus replication’ to this section. This section includes a paragraph on genetic recombination of viruses as well.

line 204, should include the following reference: Hu et al., PLoS pathogens,

2017: e1006698.

Authors’ response: done.

Lines 207-, should discuss variation of the viral loads in different season mayInitiatives also contribute the spillover. Please see the references: Wang et al., Virol Sin,

2016, 31:78-80; Lau et al., J Virol, 2010, 84: 2808-19.

Authors’ response: done.

Line 275, should included the following reference: Necker et al., PNAS, 2008,

105:19944-9.1

Authors’ response: done. Added an additional reference for the MERS-CoV reverse genetics system (Scobey et al., PNAS, 2013).

Reviewer 3 Report

The article “Bats and coronaviruses” by Banerjee and colleagues is dedicated to the relationship of bats and coronaviruses that are linked to disease in humans and agricultural animals. Within the past ~20 years, flying foxes and bats have been extensively studied as reservoir hosts of zoonotic viruses. They have been show to harbor a multitude of viruses that can cause life-threatening diseases in humans; however, bats usually do not develop symptoms. In the search of the source of the SARS pandemic, which was caused by a novel coronavirus (CoV), it was discovered that (insectivorous) bats contain diverse coronaviruses. Since then, multiple bat CoVs that are related to known CoVs of humans and agricultural animals have been discovered (at least on molecular level) and it is believed that many (if not all) CoVs that are known today have an ancestral progenitor in bats.

In this review, the authors summarize the current knowledge on the link between CoVs that are pathogenic to humans (SARS-CoV, MERS-CoV, HCoV-NL63 and HCoV-229E) or agricultural animals (PEDV and SADS-CoV) and bats. Further, the authors define factors that have been implicated to play a role for the transmission of bat-associated viruses to other species (spillover events) and give an overview about recent findings regarding the immune responses in bats that are assumed to allow bats to harbor highly pathogenic (at least for humans) viruses without developing disease. Finally, the authors combine information from all the different sub-categories in a paragraph entitled “Implications of bats as hosts of CoVs” in which they highlight areas where more information/future research is required.

This article is well-written and contains a huge load of information without overloading the reader that might not be a coronavirus aficionado. In addition, this article highlights “gaps of knowledge” that might spur future research from multiple fields (molecular biology, immunology, ecology, microbiology …). I can recommend this article for publishing after minor editing.

Minor comments:

Line 32:

On taxonomic level, Rhinolophoidea is considered a superfamily rather than a family

Line 43:

Nidovirales should be italicized

Line 44:

According to the latest release by the International Committee on Taxonomy of Viruses (ICTV), the Coronavirdae family contains two subfamilies (Alphaletovirus and Orthocoronavirinae) of which the Orthocoronavirinae harbors the alpha-, beta-, gamma- and deltacoronaviruses that have all been elevated to the genus level (Alphacoronavirus, Betacoronavirus, Gammacoronavirus, Deltacoronavirus).

Line 60:

Please delete the space between “HCoV-” and “NL63”

Line 109:

Please correct “Vespertillionidae” “Vespertilionidae”. Further, Vespertilionidae should not be italicized

Author Response

Reviewer 3:

Minor comments:

Line 32:

On taxonomic level, Rhinolophoidea is considered a superfamily rather than a

Family

Authors’ response: we have changed ‘family’ to ‘superfamily’.

Line 43:

Nidovirales should be italicized

Authors’ response: done

Line 44:

According to the latest release by the International Committee on Taxonomy of

Viruses (ICTV), the Coronavirdae family contains two subfamilies

(Alphaletovirus and Orthocoronavirinae) of which the Orthocoronavirinae

harbors the alpha-, beta-, gamma- and deltacoronaviruses that have all been

elevated to the genus level (Alphacoronavirus, Betacoronavirus,

Gammacoronavirus, Deltacoronavirus).

Authors’ response: done

Line 60:

Please delete the space between “HCoV-” and “NL63”

Authors’ response: done

Line 109:

Please correct “Vespertillionidae” “Vespertilionidae”. Further, Vespertilionidae

should not be italicized

Authors’ response: done